# Consequences and Challenges of the Fourth Industrial Revolution and the Impact on the Development of Employability Skills

**Veronika Bikse** [1], **Liva Grinevica** [2,*], **Baiba Rivza** [3] and **Peteris Rivza** [4]

1. Institute of Management Sciences (IMS), Liepaja University, 14 Liela Street, LV-3401 Liepaja, Latvia; vbikse@lu.lv
2. The Unit of Agrarian Economics Sciences, Latvian Academy of Agricultural and Forestry Sciences, 1 Akademijas Square, LV-1050 Riga, Latvia
3. Faculty of Economics and Social Development, Latvia University of Life Sciences and Technologies, 2 Liela Street, LV-3001 Jelgava, Latvia; baiba.rivza@llu.lv
4. Faculty of Information Technologies, Latvia University of Life Sciences and Technologies, 2 Liela Street, LV-3001 Jelgava, Latvia; peteris.rivza@llu.lv
* Correspondence: liva.grinevica@gmail.com

**Abstract:** The aim of this research study is to perform an analysis of the consequences and impact of the fourth industrial revolution on the development of employability skills and to identify possible solutions to help overcome these challenges. The research methodology applied in this research study involves examining and analysing literature and Internet sources. To identify possible solutions for the development of employability skills in the context of challenges of the fourth industrial revolution, the Analytic Hierarchy Process (AHP) was used. The results of the research indicate that the promotion of youth employability requires close cooperation between educational institutions and entrepreneurs, as well as that significant attention should be paid to investment in human capital and the digital transformation of business. The research expands and provides insights into the situation in Latvia regarding the impact of the fourth industrial revolution on the development of employability skills and performed an analysis of the main possible solutions for the successful development of employability skills in Latvia that can be ensured by investing in human capital and improving the business environment, the digital transformation of SMEs and the modernization of the education system. In this context, it is crucial to promote more collaboration among educators, academics, policymakers, and practitioners.

**Keywords:** Analytic Hierarchy Process (AHP); 4th industrial revolution; framework; employability competencies/skills; sustainability

## 1. Introduction

We are currently experiencing the fourth industrial revolution around the world, which is referred to as Industry 4.0, and fundamentally changes all production processes and social structures. The fourth industrial revolution is having a huge impact on society, on the processes taking place in the world, and on people's way of life, especially on the labour market. It will affect education and health systems, work, communication, self-expression, information habits, lifestyle habits and ways of traveling.

Industrial revolutions have occurred throughout history and should be seen in the context of the development of new technologies when new technologies and novel ways of perceiving the world trigger a profound change in economic systems and social structures [1].

The fourth industrial revolution is unfolding and is mostly based on the automation and robotization of production processes, which radically changes the nature of work; the simplest types of work and working functions in routines are increasingly disappearing.

They are directed to much more complex and automatable processes. All this turns human work more and more meaningful and creative, causes significant changes in the spheres of employment, and contributes to the rapid growth of highly skilled workers in production, who both know how to work with these technologies and participate in their creation themselves. Thus, it can be concluded that in the next few years, highly qualified specialists with exact knowledge and logical thinking will be needed in the labour market. At the same time, people with communication and social cooperation skills will also be needed. In addition, the nature of science and innovation is also changing significantly. For example, the use of big data, AI, interconnected networks, and high-speed computing could lead to new discoveries, create technology, and bring it into production requiring specialized human capital (such as technologists and scientists) [2].

It means that advanced technologies impose high demands on people's education, their professionalism and competences; demand is emerging for all people to build up employability and digital competences/skills in order to be able to learn and implement new technologies. It is also important to retrain existing and redundant workers. It is important to underline that, according to a survey of residents of Latvia conducted by the Marketing and Public Opinion Research Centre SKDS, businesspersons and entrepreneurs did not take sufficient care of the further education of their employees. Only 17% of respondents indicated that training opportunities were provided by their employers. At the same time, 62% of respondents, a majority, had independently tried to acquire knowledge and improve their skills in working with technologies during the last year [3]. This is indicative of the growing role of employers and education to focus on developing human capital as well as competencies and digital skills because without them one cannot implement the advanced technologies in all areas.

The fast technological changes determine the immediacy of the needed changes in higher education for responding to the new and changing world. Digital technologies are beginning to facilitate skilled human capital development in new ways, leading to a focus on the development of competent individuals as the most important precondition of competitiveness. It envisages implementing competence-based education. This requires a complex approach not only to tackling organizational problems in the learning/training process, placing the main focus on the formation of personality and creating opportunities for self-realization, but also to targeted education management. In addition, there is the need to implement studies remotely by carrying out a broader digitalization of the learning process, which is an essential investment in the future, so that the younger generation can gain invaluable experience in how to do things differently and more efficiently in everyday life using technologies [4].

In this context, according to the Digital Economy and Society Index DESI 2021, an important research problem is that in the EU, the level of digital skills has continued to grow slowly. In 2019, the percentage of people in the EU that had at least basic digital skills reached 56% (up from 55% in 2017) and 31% with above-basic digital skills; 58% of individuals had at least basic software skills [5]. The results of the present research are considerably consistent with the survey results of 289 Baltic CFOs of the largest SEB Baltic countries, which show that the shortage of skilled professionals is the common top issue for the companies in all three Baltic countries: 59% in Lithuania, 54% in Latvia and 46% in Estonia. However, the survey's results confirm that the top priority for Baltic companies is the focus on digitalization, automation and innovation for the next year [6].

This means that the main problem in Latvia with regard to the implementation of advanced technologies was a lack of qualified specialists and the fact that a large segment of the population lacked even basic digital skills. Thus, policy makers are facing major challenges when identifying the envisaged changes and making appropriate adjustments to the systems of education and retraining of the unemployed [7]. Therefore, the aim of this research study is to perform an analysis of the consequences and the impact of the fourth industrial revolution on the development of employability skills and to identify possible solutions to help overcome these challenges.

The questions addressed in this paper are as follows:

What are the consequences and challenges of the fourth industrial revolution and the impact on the labour market?
What competences/skills will be needed in the future because of technological change?
What is the level of digitalization of businesses and an adequate competence/skill of human capital in Latvia?
What are the possible solutions for successful development of employability skills in Latvia?

## 2. Research Methodology and Participants

Employability in the context of the 4th industrial revolution and the integration of young generations into the labour market is a dynamic and complex system. Equally complicated methods should be used for research on it, including both qualitative and quantitative indicators for analysis. To achieve the aim of the present research study the specialized literature, policy-related papers, Internet sources and statistical data was examining, analysing and summarizing, and theoretical framework of 21st century skills for youth employability support is elaborated.

To get a deeper insight into the problem and to identify the possible solutions for development of employability skills in the context of challenges of the 4th industrial revolution, an interview and an analysis of the opinions were carried out as the key method of the research. In total, four experts were interviewed. The experts were asked to make their evaluations of 4 possible scenarios by using the Analytic Hierarchy Process (AHP) method. The selection of experts first involved expert groups representing employers and employees, municipalities and educational institutions. One the most prominent representatives with experience was then selected from each group to represent the group's views. The selection of experts was determined by the condition that they were involved in the integration of young people into the labour market and represent the spatial levels of the labour market. The identities of the experts were confidential.

In order to organize the experts' work scientifically and correctly and to process the results of the experts' evaluations, the authors used the AHP and the key principles of the expert method [8,9].

The AHP decision-making method has been esteemed by many scholars as a method that improves the quality of decision making. We can conclude that the AHP is an efficient method for multi-criteria decision making. Decision making considers numerous different criteria. The AHP realizes a comparative analysis of the criteria that impact decision making. In this manner, it determines the best available alternative. The AHP is considered as a new approach that ensures fast, simple and rational decision-making [10].

Basically, the AHP helps in structuring the complexity, measurement, and synthesis of rankings. The features make it suitable for a wide variety of applications. The AHP has proved a theoretically sound and market tested and accepted methodology. It is almost universally adopted as a new paradigm for decision making coupled with its ease of implementation and understanding, which constitute its success. More than that, it has proved to be a methodology capable of producing results that are consistent with perceptions and expectations. Saaty describes the seven pillars of the AHP as follows:

- Ratio scales, proportionality and normalized ratio scales;
- Reciprocal paired comparisons;
- The sensitivity of the principal right eigenvector;
- Clustering and using pivots to extend the scale;

   Synthesis to create a one-dimensional ratio scale for representing the overall outcome;

- Rank preservation and reversal;
- Integrating group judgements [11].

The mathematical data processing was performed by means of the computer program Ms Excel. The criteria comparison results were illustrated in a graphic figure as minimal, maximal and average values of each factor's priority vector. A differentiated approach was

used for scenario building and evaluation. The researchers discussed and jointly decided with the experts' criteria and scenarios for the AHP hierarchy scheme. The development of the scenarios was based on the literature review, experience in other EU Member States and an in-depth examination of the situation in Latvia.

It is important to draw attention to these challenges and to find answers to the questions:

- Which of the scenarios for developing employability skills in youth can be implemented in Latvia in the near future?
- Which of the scenarios is the most appropriate for the development of youth employability skills and the integration of youth into the labour market?
- Which of the scenarios are more focused on the interests of all stakeholders involved?
- Which scenario will ensure the economic development of the country?

Four possible scenarios for developing employability skills in youth and the integration of the young people into the labour market in the context of the challenges of the 4th Industrial Revolution were evaluated by the experts. The experts were asked to assess the criteria for each scenario by applying hierarchy analysis.

The research study involved four experts who were asked to make their evaluations of the above scenarios: a school director, a deputy chairman of the city Council for Education, a professor/entrepreneur and a young individual (Table 1). The selection of experts was determined by the condition that they were involved in the integration of young people into the labour market and represented the spatial levels of the labour market. The identities of the experts were confidential.

**Table 1.** Information on the experts who participated in the hierarchy analysis.

| Expert's | | Spatial Level of Activity |
|---|---|---|
| Code | Position | |
| A | School director | Education institution interests |
| B | Deputy chairman of the city Council for Education | Local government interests |
| C | Professor/ Entrepreneur | Higher educational interests/Entrepreneurs' interests |
| D | Young people | Individuals' (in this case, youth) interests |

Source: authors' survey results.

## 3. The Fourth Industrial Revolution: The Opportunities, Challenges and Consequences, the Potential Impact on the Labour Market

The fourth industrial revolution should be seen in the context of technical achievements with major effects on the economy and in accord with the characteristic stages of the Industrial Revolution from a historical perspective to the present day. Originally, the first industrial revolution occurred from 1760 to 1840, and was launched by the development of the steam engine, the mechanization of textile manufacture and the use of coke instead of charcoal, followed by the mass production of steel and lastly, the development of railways. Thus, it changed our lives and the economy from an agrarian and handicraft economy to one dominated by industry and machine manufacturing [12,13]. However, the second industrial revolution (from 1870 to 1914) used electric power for the mass production of steel, electrification, telecommunications, and lastly the development of the motor car and the production line. The third industrial revolution began in the middle of the last century with the development of digital systems, communication, and rapid advances in computing power, which have enabled new ways of generating, processing and sharing information. Thus, electronics and information technology were used to automate production [12,14]. The literature review shows that some authors (Davis, Prisecaru, Rabana, Tregenna) have made a summary of the main features of the industrial revolutions based on some experts' opinions at Davos World Economic Forum in 2016 [15]. Our position on the main features

identified by various authors could be supported in general. At the same time, the authors consider that a more detailed description of the main features of the industrial revolutions in the period 1760–2015 is given by Prisecaru (Table 2). After summarizing the findings, he concluded that whether it is or is not the third or fourth industrial revolution, this new cycle is based on the Internet and green energies, with the former allowing easy access to information and easy trade in goods and services and the latter diminishing the energy impact on the environment [16].

**Table 2.** Main characteristics of the industrial revolutions.

| Period | Transition Period | Energy Resource | Main Technical Achievement | Main Developed Industries | Transport Means |
|---|---|---|---|---|---|
| I: 1760–1900 | 1860–1900 | Coal | Steam Engine | Texile, Steel | Train |
| II: 1900–1960 | 1980–1960 | Oil, Electricity | Internal Combustion Engine | Metallurgy, Auto, Machine Building | Train, Car |
| III: 1960–2000 | 1980–2000 | Nuclear Energy, Natural Gas | Computers, Robots | Auto, Chemistry | Car, Plane |
| IV: 2000- | 2000–2010 | Green Energies | Internet, 3D Printer, Genetic Engineering | High Tech Industries | Electrine Car, Ultra-Fact Train |

Source. Prisecaru, [16].

Now, the fourth industrial revolution is building on the third one, and it is characterized by a fusion of technologies that is blurring the lines between the physical, digital, and biological spheres. For example, engineers, designers, and architects are combining computational design, additive manufacturing, materials engineering, and synthetic biology to pioneer a symbiosis between microorganisms, our bodies, the products we consume, and even the buildings we inhabit [17]. This makes it possible to integrate different scientific and technical disciplines. For better results, people can work in teams and from different countries of the world to create new markets and new growth opportunities for each participant in the innovation. According to Montresor, the fourth industrial revolution is being driven by a staggering range of new technologies that are blurring the boundaries between people, the Internet and the physical world. It is a transformation in the way we live, work and relate to one another in the coming years, affecting entire industries and economies [18].

The literature review shows that there are various aspects of the opportunities of the fourth industrial revolution which cover wide-ranging fields, such as artificial intelligence (AI), cloud computing, advanced robotics, the Internet of Things (IoT), autonomous vehicles, 3D printing, nanotechnology, biotechnology, materials science, energy storage and quantum computing and other emerging technologies in a way that is more profound than previous waves of change, such as the one driven by the microprocessor revolution of the 1970s [17,19].

The major features of the four industrial revolutions can be characterized by the following:

- First, radically different technologies for manufacturing and services have been created: a digital economy, automation and robotization.
- Second, impressive progress has been made in artificial intelligence (AI) in recent years, driven by exponential increases in computing power and by the availability of vast amounts of data. AI-based systems can be purely software-based, acting in the virtual world, or embedded in hardware devices. Artificial intelligence is already all around us, from self-driving cars and drones to virtual assistants and software that translate or invest. Engineers, designers, and architects are combining computational design, additive manufacturing, materials engineering, and synthetic biology to pioneer a symbiosis between microorganisms, our bodies, the products we consume, and even

the buildings we inhabit [17,20]. The artificial intelligence (AI) provides new forms of work and organization.

- Third, new technologies have made possible to use a new generation of autonomous robots. Equipped with cutting-edge software, AI, sensors and machine vision, these robots are capable of performing difficult and delicate tasks, and can recognize, analyse, and act on information they receive from their surroundings [21]. Advanced robotics have the potential to create new types of jobs, improve and change the quality of people's lives in the near future. The research studies done by the McKinsey Global Institute revealed that between 2016 and 2030, the impact of automation change in hours worked by 2030 for the United States is expected to increase by 60% and for 14 Western European countries by 52%, and the highest demand will be observed for advanced IT and programming skills, as well as for basic digital skills, which could grow as much as 90 and 70 percent between 2016 and 2030. The demand for other skills that constitute this category will also grow, but not as strongly [22].

- Fourth, impressive progress has been made by using 3D printing, which brings together computational design, manufacturing, materials engineering and synthetic biology, reduces the gap between makers and users and removes the limitations of mass production [21]. In addition, 3D printing allows entrepreneurs with new ideas to establish small companies with lower start-up costs. The entrepreneur can bring the product with 3D printing, without the traditional time constraints but with a broader range of applications, from mass customization to distributed manufacturing [13,21].

- Fifth, the Internet of Things (IoT) is an appropriate solution for combining any type of object into a single digital network through which the connectivity of modern devices, systems and services takes place, which will provide automation in almost all areas. In this way, devices and parts are integrated into the virtual environment in a real environment [18,23]. With the development of the Internet of Things (IoT), the number of everyday objects endowed with "smart" functions and able to seamlessly perform health monitoring and notice early signs of the disease is increasing. For example, a smartwatch worn on the hand is able to monitor the heart rate and other health parameters [24].

- Sixth, Big Data are collected to obtain more information from a wide range of sources, from factory equipment and Internet of Things (IoT) devices, to ERP and CRM systems, to weather and traffic apps [21]. The value of the use of big data in the production process is based on the fact that the data can be used in the automation, visualization and analysis of industrial processes.

- Seventh, Cloud computing is a combination of Internet services that connect information resources and software to different servers, enabling users to access data from different locations. Cloud solutions have already become an integral part of the daily concern of many companies. Soon, large companies will start giving up their server space and migrate to secure cloud storage to make it easier for employees to access data and optimize costs.

- Eighth, A blockchain is essentially a network of computers and one of the key emerging technologies that can help to make interaction between individuals, enterprises and public organisations more efficient, reinforce trust and enable each party to retain control of their own data. Blockchain technology has been mainly linked to financial services and cryptocurrencies, and it is now expanding into other sectors, such as media and telecommunications, healthcare and government services [20,25]. According to Schwab (2017), 10% of global gross domestic product (GDP) is based on blockchain technology [26].

The opportunities of the fourth industrial revolution are incontrovertible. According to Schwab [1], in this revolution, emerging technologies and broad-based innovation are diffusing much faster and more widely than in previous ones, which continue to unfold in some parts of the world. The fourth revolution is distinguished from the third revolution by velocity, breadth, depth and systems impact. The fourth industrial revolution is a fun-

damental shift in the way technology, communications, data, and analytics impact almost all aspects of society and the economy: businesses, governments and people. Disruptive technologies offer great opportunities for growth given their transformational impact on companies' products, processes and business models, as well as on the public sector. It will affect people's identity, sense of privacy, ownership, consumption patterns, work and leisure, careers and skills [1,17,20,26,27].

History indicates that consumers have gained the most from industrial revolutions, as a lot of activities can now be carried out remotely. For example, organizing transport, booking restaurants, buying groceries and other goods, making payments, listening to music, reading books or watching films—these tasks can now be done instantly, at any time and in almost any place [14]. In the future, technological innovation will also lead to a supply-side miracle, with long-term gains in efficiency and productivity. Transportation and communication costs will drop, logistics and global supply chains will become more effective, and the cost of trade will diminish, all of which will open new markets and drive economic growth [1,17].

Throughout the ages, technological innovations have tended to raise labour productivity by replacing existing workers with technology, and they also usher in new products and processes that open up new sources of growth [28]. In digitalized, robotic and automated factories, much of the work is performed by equipment and robots, and humans only monitor them—this is the reality of many companies in the world. While there are many benefits of the fourth industrial revolution, there are several key challenges that lie ahead. Automation and robotics technologies take over the simpler, more routine tasks, and now threaten many professions, such as accountants, taxi drivers and paralegals. Many of the jobs that have disappeared might not come back [29]. Moreover, technological advancements could yield greater inequality [30]. However, studies show that the fourth industrial revolution will lead to far more jobs being created than will be lost. For example, a study conducted in Austria found that around two-thirds of jobs are at risk in low-skilled occupations. At the same time, in sectors with a higher proportion of digitalisation, 390,000 jobs were created and 75,000 were lost. In sectors with a lower proportion of digitalisation, 189,000 jobs were created and 280,000 were lost. This study shows a positive correlation between employment growth and new jobs creation due to the progress of digitalisation [31]. Similarly to previous studies, Zahidi [32] proposed four compelling predictions:

1. The production process is automated faster than expected, displacing 85 million jobs in the next five years.
2. The robot revolution will create 97 million new jobs.
3. In 2025, analytical thinking, creativity, and flexibility will be among the most sought-after skills.
4. The most competitive businesses will focus on upgrading their workers' skills.

Thus, today it is obvious that due to technological innovations, labour market changes can take place quite rapidly, and under such circumstances not everyone has been able to benefit. Low-skilled jobs will be replaced by automation technologies. Furthermore, the fourth industrial revolution will affect all professions and all groups of people, especially youth. According to Kassid, many jobs will disappear, and the new jobs that are created will leave many young people behind [33]. Those with lower skills will join the swelling numbers of youth not in education, employment or training (NEET) or will find themselves in insecure jobs, with lower-paid working conditions. Additionally, a situation may arise where both high demand for new employees and high unemployment exist simultaneously, as in-demand skills differ from those available on the labour market. This means that employees will have to adapt accordingly by learning and applying new knowledge and skills. Within the context of this, it is necessary to seek answers to the questions on what competencies and skills may be needed in the future according to technological change and the challenges of the labour market.

## 4. Competencies and Skills Needed in the Future According to Technological Change

According to Debczak, the 21st century will face the challenges of the fourth industrial revolution and mostly three generations will dominate the labour market; the first, Y or the Generation of Millennials, were born between 1981 and 1996 (25–40 years old). It is a generation that has grown up with technology, as it has been shaped by a technological revolution and rapid technological progress. The second is Generation Z—anyone born between 1997 and 2012 (9–24 years old) [34]. It is today's youth, called the online generation that has been totally immersed in the world of the internet since birth. Their lifestyles are expected to be shaped by the high-tech era and the widespread use of social media. A research study on Gen Z and Millennials conducted by the entertainment media Whistle based in the USA and the United Kingdom in autumn 2020 to decode their relationships to podcasts revealed that 75% of pod users have listened with the goal of learning something new. However, 67% of people aged 13–34, in search of new knowledge, preferred podcasts over books [35]. Generation Alpha, representing anyone born between 2010 and 2025 (0–11 years old) is characterised by online children who feel comfortable in the digital world, using a variety of applications on mobile devices that the middle generation might not understand, because it is another world that has not been experienced.

Taking into account the fact that in the future, more challenges of the fourth industrial revolution will be faced by young generations, the labour market is expected to set much higher requirements for them with regard to both career readiness and personal traits, i.e., the development of employability. It means that to build up employability and successfully compete in the labour market in the 21st century, the youth have to meet high labour market requirements and face professional challenges.

It has to be noted that in recent years, the concept of employability is increasingly used together with another concept—employment. It is important to underline that the concepts of employability and employment are not the same thing. The research findings show that the concept of employability is much broader than that of employment. Little considers that employability is a multi-dimensional concept, and there is a need to distinguish between factors relevant to obtaining a job and factors relevant to the preparation for work [36]. Being employed means having a job, but being employable means having the qualities needed to maintain employment and progress in the workplace. It means that employability is a prerequisite for employment [37,38]. Employability is not just about preparing youth for employment. It is about supporting students to develop a range of knowledge, skills, behaviours, attributes and attitudes which will enable them to be successful not only in employment but also in life [39]. Talin explains the most important 23 skills of the future and divides them into two different areas: the basis and foundations which are necessary for work in the future, the job-specific skills and competencies to approach complex problems, and personal qualities to improve over time to adapt as smoothly as possible to changes [40]. According to Robinson, employability are those basic skills necessary for getting, keeping and doing well on a job. He suggested to divide employability skills into three skill sets: basic academic skills, higher order thinking skills, personal qualities [41]. However, Stiwne & Alves believe that ka employability is a combination of four different aspects such as knowledge regarding the subject, skilful practices or procedural knowledge, efficacy beliefs and meta cognition [42].

A wider description of employability skills is given by other authors. For example, Awofala et al., consider the teachable skills which are required to get, keep and do better at the workplace and which are also referred to as core skills, basic skills, transferable skills, generic skills, key skills, soft skills, behavioural competencies skills and cross-curricular skills [43]. McGunagle and Zizka consider that the employability skills of the 21st century include a wide range, e.g., ICT literacy, problem solving, critical thinking, innovation, decision making, creativity, collaboration, and information literacy [44]. Van Laar et al., based on a recent systematic literature review of academic literature, have identified 21st century skills and digital skills and proposed seven core skills which are fundamental for performing tasks in a broad range of occupations (technical, information manage-

ment, communication, collaboration, creativity, critical thinking, and problem solving) and five contextual skills (ethical awareness, cultural awareness, flexibility, self-direction and lifelong learning) [45,46]. They point out that the dynamic changes in the types of jobs demanded by the knowledge society pose serious challenges to educational systems, as they are currently asked to prepare young people for jobs that might not yet exist.

Based on the literature review, the authors summarize some of the main competencies and skills which strongly relate to employability and are needed in the future according to technological change and the challenges of the labour market (Figure 1).

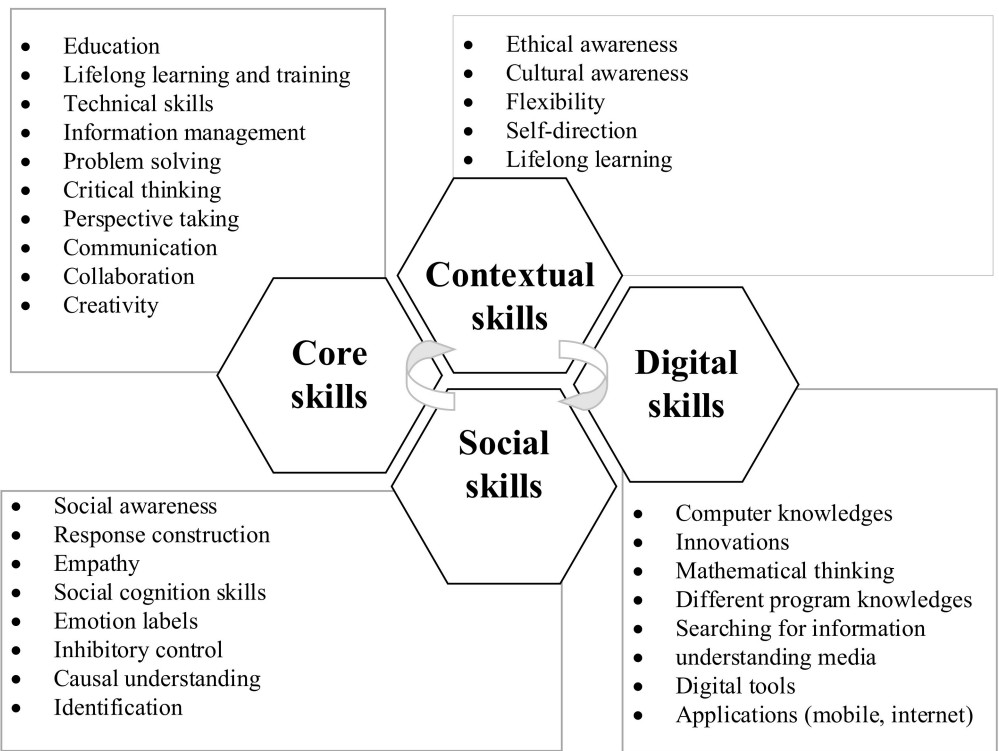

**Figure 1.** Framework with 21st century competencies and skills for employability support. Source: authors' compilation.

As shown in Figure 1, employability skills which are needed in the 21st century include such main elements as digital skills, social skills, core skills and contextual skills. In addition to the competencies and skills needed in the 21st century that are shown in Figure 1, the Commission on the Futures of Education has mentioned that future skills also include scientific literacy, with the remark that scientific literacy does not conflict with human values. Science has become part of human daily life today—we use scientific achievements and inventions every day, and it is expected that in the future the importance of scientific literacy of society will only increase. The report has stressed that research and innovation enable us to systematically learn together—to reflect, to experiment and have an impact on society together and, in doing so, to reimagine our futures together. The development of research, innovative, digital, entrepreneurial and, globally, civic and other competences in interaction with professional competences helps a person to become a high qualified professional, as well as contributing to individuals' understanding of the commonality of society and their role in it [47].

In the specialized literature, policy-related papers and reports, several different approaches to the concept of employability, competences and skills needed in the 21st century are suggested. After having studied them, the conclusion is that, except for a few differences, all the authors focus more or less on the same issue and could be supported in general. At the same time, the authors consider that employability is a combination of several factors building up the young individual's key competencies and different skills,

including scientific literacy. The demand of today's labour market is different and changing very rapidly, so the most important thing is to be able to adapt and keep up with the times, acquiring new employability skills and abilities that will be needed in the future labour market situation. The key word representing the fourth industrial revolution is adaptability.

## 5. Hierarchy Analysis of the Development Scenarios of Young Generation Employability in the Context of Challenges of the Fourth Industrial Revolution

Youth employability is affected by a number of factors: the education system, career services, external conditions, labour market conditions, as well as government policy measures aimed at promoting employability [48]. A holistic approach is required to ensure the implementation of all the above-mentioned factors and activities. The present research focuses on an analysis of the education system, employers and government policies aimed at developing youth employability skills in the context of the 4th industrial revolution. These factors could well serve as a starting point to investigate the youth's employability for using the Analytic Hierarchy Process (AHP) and the key principles of the expert method.

Based on the expert interview results, scenarios and criteria were selected. The experts chose the most important criteria groups for young generation employability in the context of challenges of the fourth industrial revolution.

According to the expert interview results, there were defined the following criteria groups:

- Individual growth.
- Social inclusion.
- Welfare raising.
- Investment attraction.
- Competitiveness.
- Sustainable development.

The criteria hierarchy for improving the integration of young people into the labour market in the context of the fourth industrial revolution was created after discussions with the experts.

Four possible scenarios for improving the integration of young people into the labour market in the context of the fourth industrial revolution were offered for expert evaluation:

The first scenario is: "Promoting the development of human capital and the business environment". It involves attracting financial resources for investments in human capital to promote start-ups and self-employment of young people as well as providing for the establishment of a special financial and administrative support program. It would be important to involve young people in entrepreneurship, especially by fostering the development of entrepreneurship in the context of digitalization and by encouraging the creation of start-ups in competitive sectors that will continue to be in demand. It is therefore important to stimulate young people's interest in developing and promoting entrepreneurship.

The second scenario is: "Cooperation between educational institutions and entrepreneurs for development of employability". This constitutes the involvement of entrepreneurs in the training of highly qualified and competitive specialists. It is important for entrepreneurs to work with professional and higher education institutions to ensure that young people are prepared for employability and to facilitate their integration into the labour market by setting up a professional training system in a real working environment.

The third scenario is: "The role of policy makers, educational institutions and entrepreneurs in lifelong learning". Tax incentives for entrepreneurs employing young people would have a positive effect on promoting youth employment. By ensuring successful interaction between the state, its institutions, the amount of funding and human capital, it is possible to improve the country's economic employment situation by promoting the effective integration of young people into the labour market and their future competitiveness.

The fourth scenario is: "The EU's role in the development of young generations' competencies and skills for growth". The integration of young people into the labour

market in the context of the fourth industrial revolution would have a positive impact on the effective use of EU structural funds in youth mobility activities organized by the State Employment Agency (SEA). It is important to draw young people's attention to business opportunities and inform them about business support programs, professional development in various courses and internships in companies, where young people often continue their work after participating in SEA programs (Figure 2).

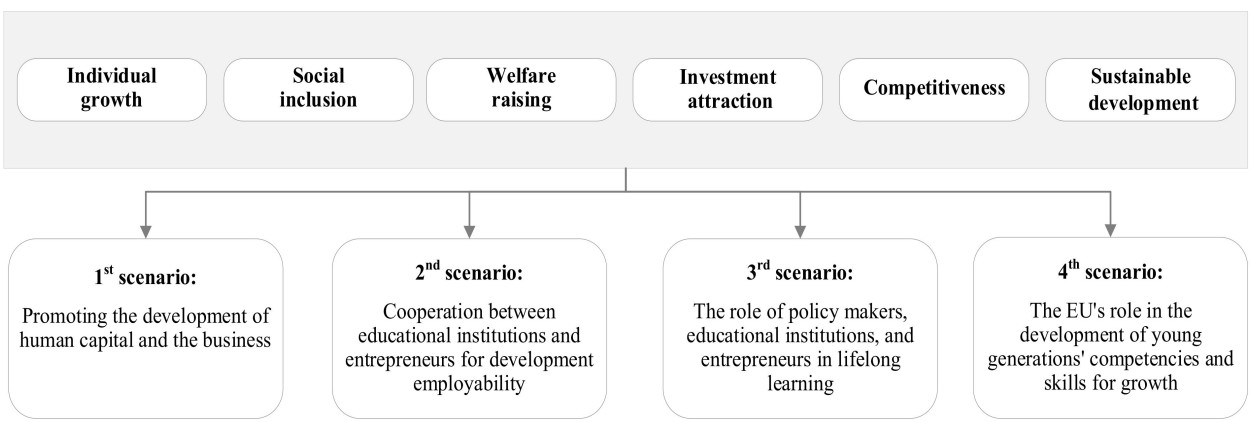

**Figure 2.** Criteria hierarchy for "Improving the integration of young people into the labour market in the conditions of the 4th industrial revolution". Source: authors' construction.

According to the experts, the most important criteria were "Individual Growth" (0.29) and "Sustainable development" (0.22). "Competitiveness", "Raising welfare" and "Social inclusion" were rated similarly (Figure 3).

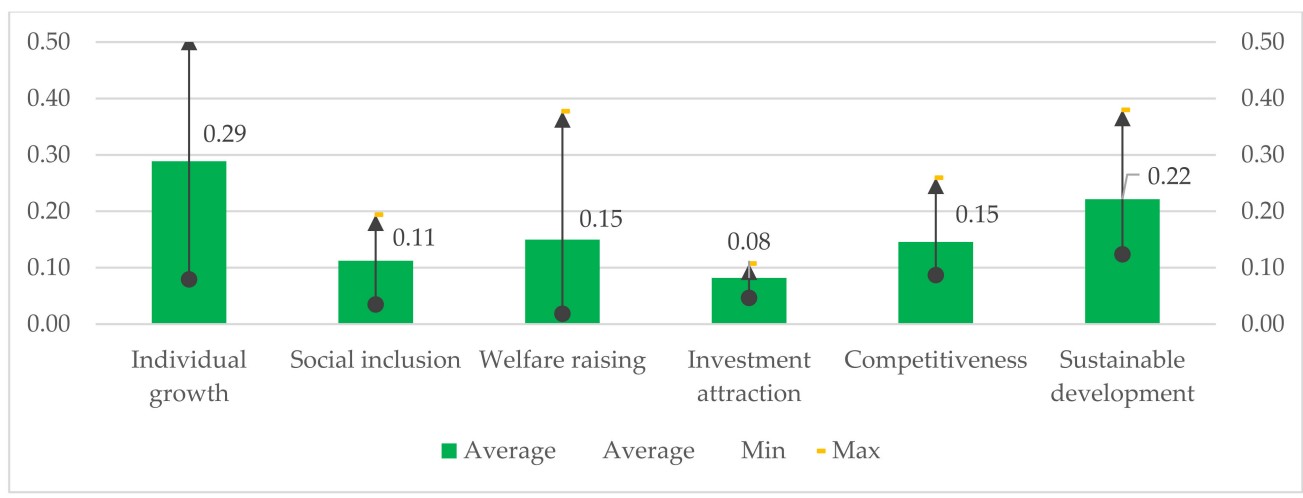

**Figure 3.** Ratings of the criteria groups. Source: authors' survey results.

The greatest disagreement among the experts was over the criteria "Individual growth" and "Raising welfare" (Figure 3). The AHP methodology and a 9-point scale were used for evaluation.

The experts evaluated all four scenarios according to each of the six criteria. Let us look at the results of the evaluation. In relation to the criterion of "Individual growth",

the second scenario "Cooperation between educational institutions and entrepreneurs for development employability" (0.50) was very convincingly superior, followed by the scenario "The role of policy makers, educational institutions, and entrepreneurs in lifelong learning" (0.26) lagging by almost half. It should be noted that the ratings of the scenarios according to this criterion revealed that the experts were strongly united, which was indicated by the small scatter of their ratings (Figure 4).

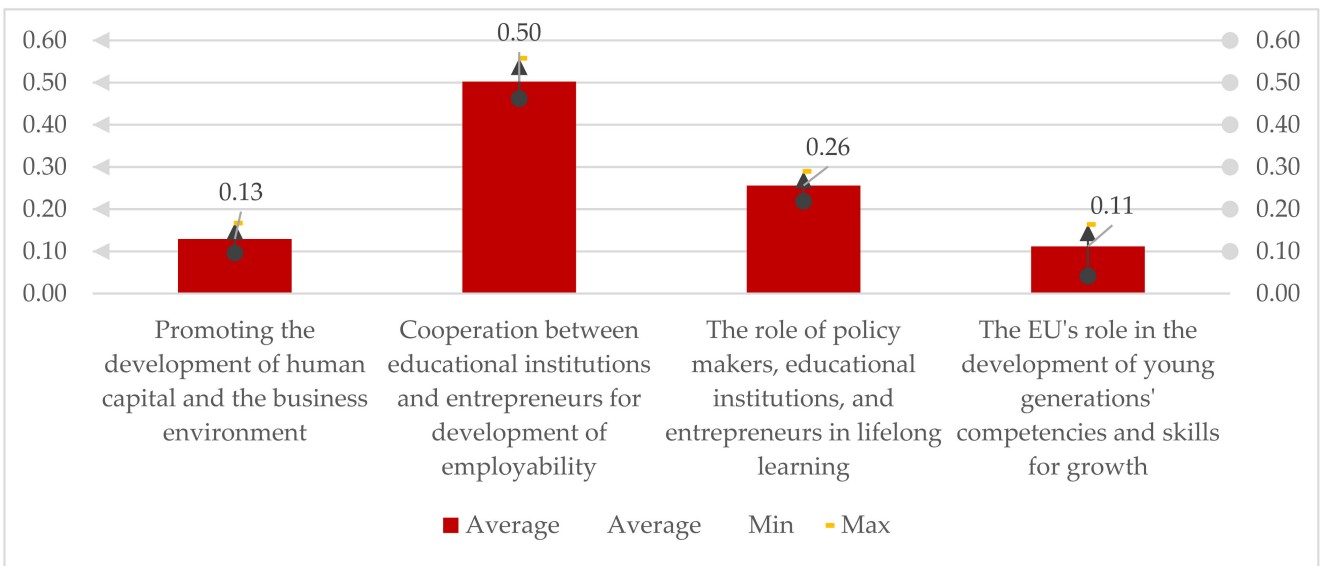

**Figure 4.** Ratings of the scenarios in terms of individual growth. Source: authors' survey results.

The ratings of the scenarios according to the criterion "Social inclusion" were also similar, namely, the second scenario "Cooperation between educational institutions and entrepreneurs for development of employability" was rated the highest (0.46), followed by the scenario "The EU's role in the development of young generations' competencies and skills for growth", which lagged far behind (0.23). However, it should be noted that in this case, the opinions of the experts differed significantly (Figure 5).

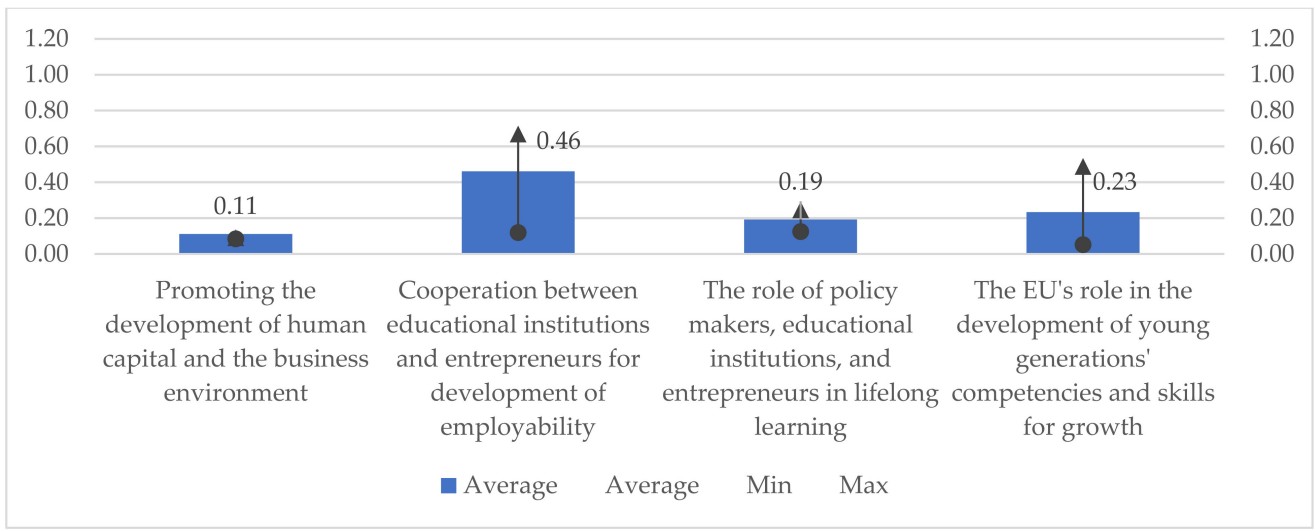

**Figure 5.** Ratings of the scenarios in terms of social inclusion. Source: authors' survey results.

The ratings of the scenarios according to the criterion "Welfare raising" differed, and the best scenario was Scenario 3 "The role of policy makers, educational institutions, and entrepreneurs in lifelong learning" (0.39), followed by "The EU's role in the development

of young generations' competencies and skills for growth" (0.30), although the experts were not in agreement (Figure 6).

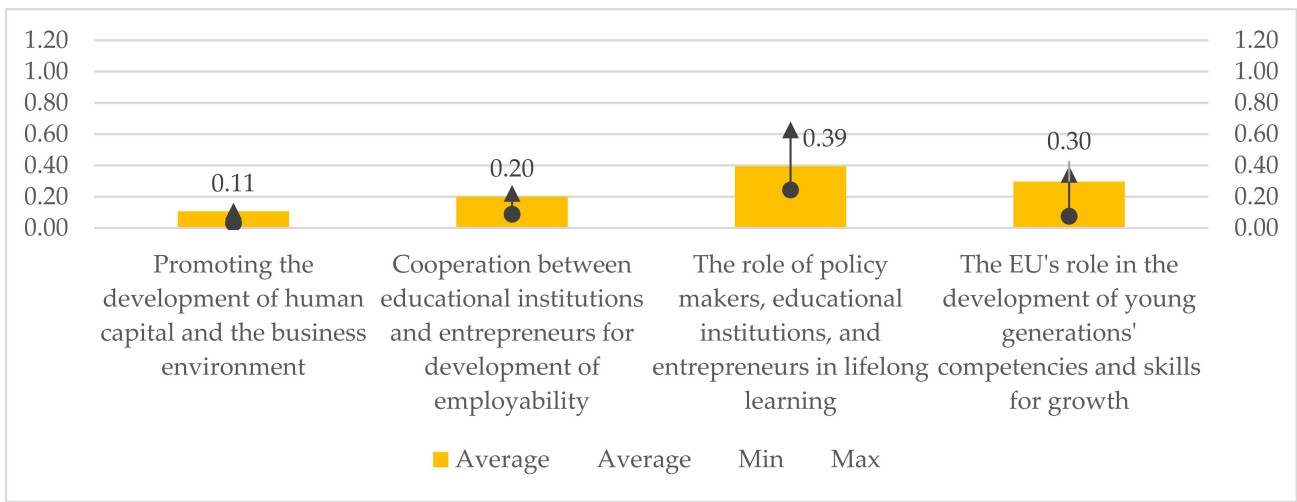

**Figure 6.** Ratings of the scenarios in terms of welfare raising. Source: authors' survey results.

The situation was similar if the scenarios were evaluated according to the criterion "Investment attraction"—the best scenario was Scenario 3 "The role of policy makers, educational institutions, and entrepreneurs in lifelong learning" (0.34), followed by "The EU's role in the development of young generations' competencies and skills for growth" (0.32), although experts did not have unanimous opinions on it (Figure 7).

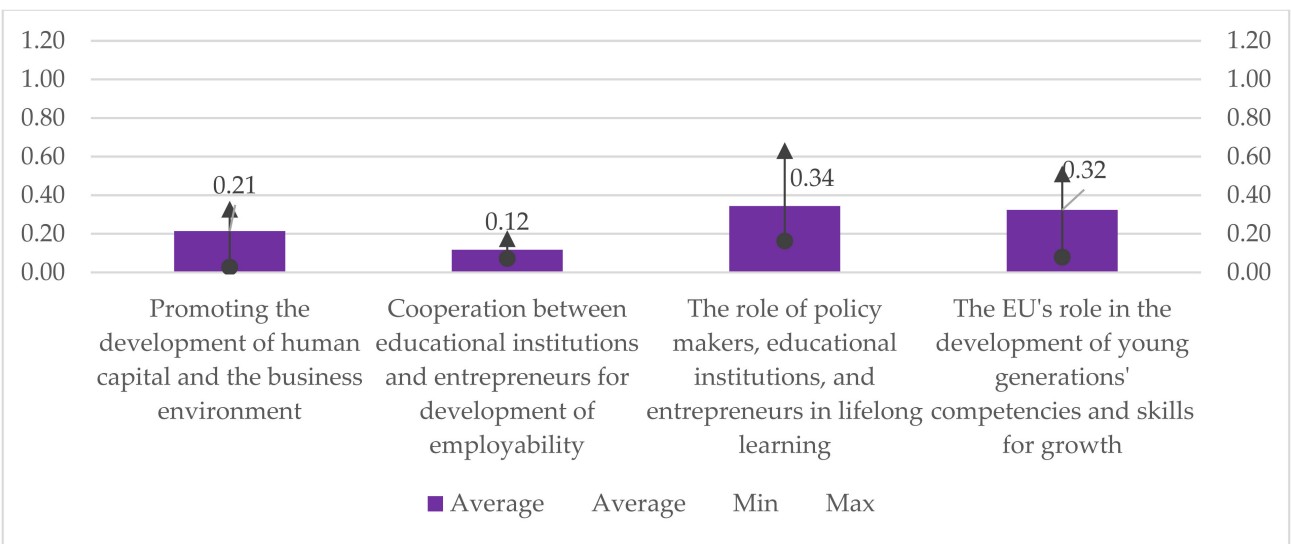

**Figure 7.** Ratings of the scenarios in terms of investment attraction. Source: authors' survey results.

In addition, the ratings of the scenarios according to the criterion "Competitiveness" revealed that the best scenario was Scenario 3 "The role of policy makers, educational institutions, and entrepreneurs in lifelong learning" (0.37), followed by the scenario "Cooperation between educational institutions and entrepreneurs for development of employability" (0.25). The opinions of the experts differed more on Scenario 3 and Scenario 4 (Figure 8).

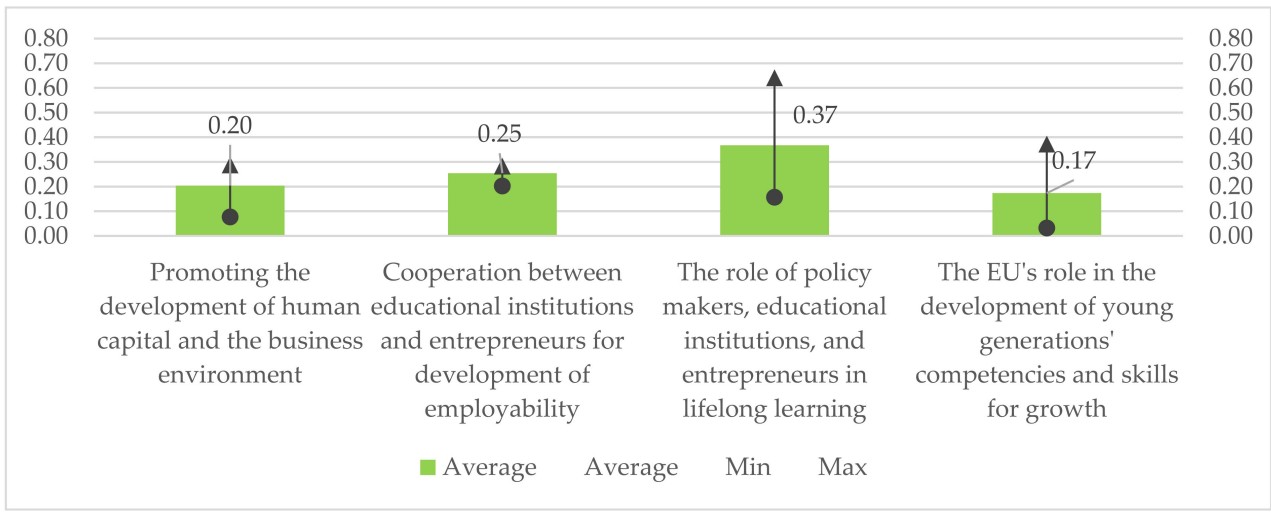

**Figure 8.** Ratings of the scenarios in terms of competitiveness. Source: authors' survey results.

The ratings of the scenarios according to the criterion "Sustainable development" were similar to those according to the criterion "Individual growth"; the second scenario "Cooperation between educational institutions and entrepreneurs for development of employability" (0.52) was convincingly superior, followed by the scenario "The role of policy makers, educational institutions, and entrepreneurs in lifelong learning" (0.21) (see Figure 8). It should be noted that the ratings of the scenarios according to this criterion revealed that the experts were not very united, which was indicated by the large scatter of their ratings (Figure 9).

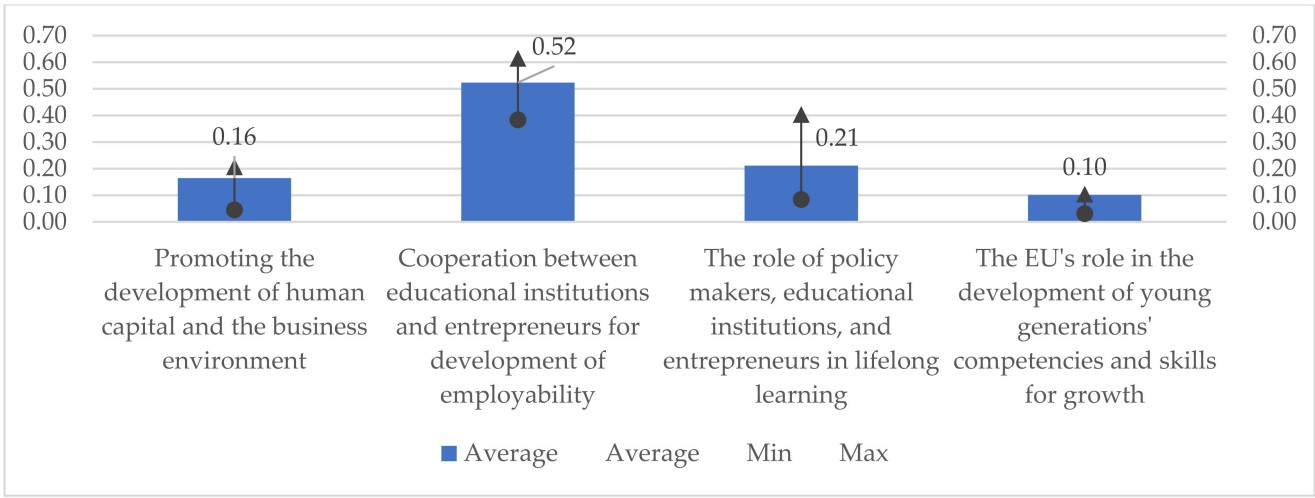

**Figure 9.** Ratings of the scenarios in terms of sustainable development. Source: authors' survey results.

After summarizing the ratings of the scenario according to all six criteria, the leader was the second scenario "Cooperation between educational institutions and entrepreneurs for development of employability" (0.39), the second best scenario was "The role of policy makers, educational institutions, and entrepreneurs in lifelong learning" (0.28), followed by "The EU's role in the development of young generations' competencies and skills for growth" (0.17) and finally, the scenario "Promoting the development of human capital and the business environment" (0.16) (Figure 10).

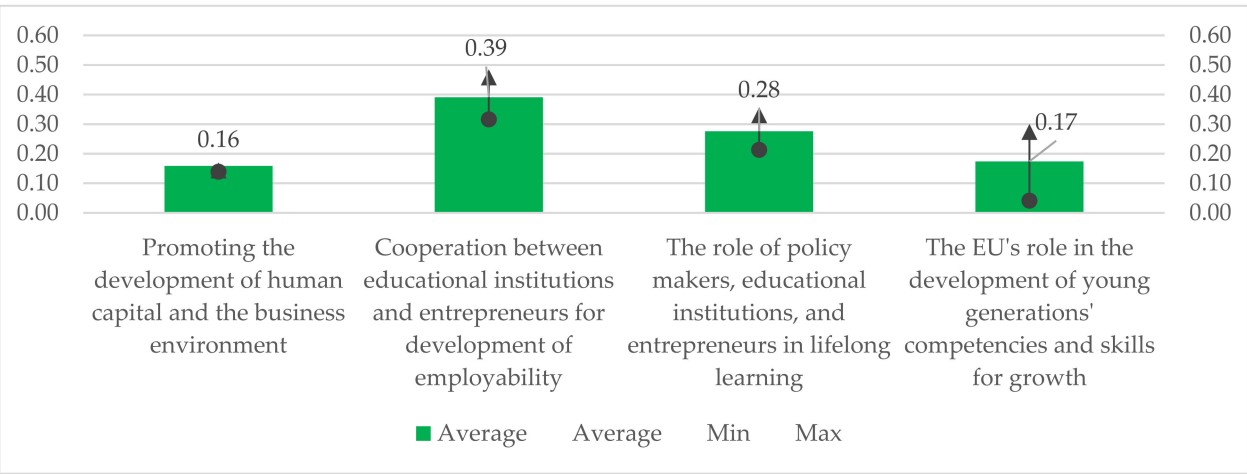

**Figure 10.** Summary ratings of the scenarios according to all criteria. Source: authors' survey results.

The AHP method measures the consistency of expert opinions by the consistency ratio (C.R.), which must be less than 0.2 [8,9]. For all the experts and at all the levels of the hierarchy, the C.R. ranged from 0.01 to 0.19.

According to the research results, complex approaches must be applied to improving the integration of young people into the labour market in the context of the fourth industrial revolution.

## 6. Discussion. Instruments for Successful Development of Employability Skills and Transition to the Labour Market in the Context of Challenges of the 4th Industrial Revolution

### 6.1. What Is the Level of Digitalization of Businesses and an Adequate Competence/Skill of Human Capital in Latvia?

The results of the present research are considerably consistent with the DESI 2021, which showed that the development of human capital and the implementation of digital technologies across sectors in Latvia lagged behind EU averages [5]. According to the human capital dimension of the DESI 2021, Latvia was ranked 20th among the EU-27 Member States, which was much lower than the EU average. In terms of digital skills, only 43% of the population aged 16–74 have at least basic digital skills, compared with an average of 56% in the EU. However, specialists in the field of information and communication technologies made up only 3.7% of the total number of employees in Latvia, compared with an average of 4.3% in the EU. A lack of digital skills is a major barrier to the wider use of digital solutions in the private sector.

According to the DESI 2021, Latvia had one of the lowest scores on the index of integration of digital technology of businesses and e-commerce compared with the EU average (Figure 11).

It is important to note that in Latvia, significant progress has been made in the digital transformation of some areas in recent years. For example, according to the DESI 2021, Latvia has a leading position in terms of broadband coverage and usage, and is well prepared for the introduction of 5G. The country's main strengths are its high-speed broadband (NGA) coverage (93% compared with an EU average of 87%) and 39% households subscribe to at least 100 Mbps high-speed broadband, compared with an EU average of 34%. Latvia has almost full 4G coverage (99.9%) and has already allocated the 5G radio spectrum. In addition, in terms of digital public services, the performance of Latvia across all categories is higher than the EU average.

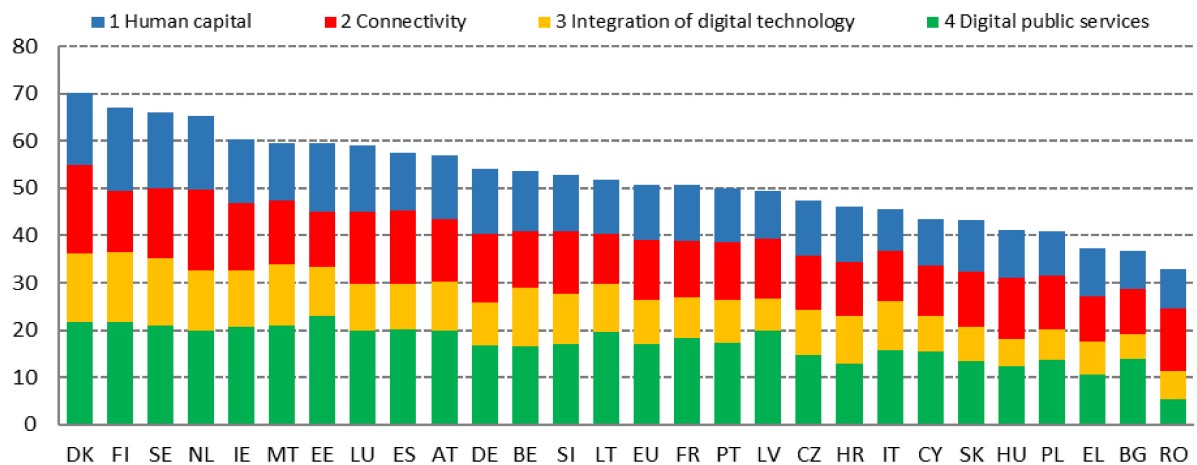

**Figure 11.** Digital Economy and Society Index, 2021. Source: [5].

A study on the sustainability of SMEs in terms of digital transformation shows that cloud computing such as Google Drive, iCloud and Dropbox and social media such as LinkedIn and Facebook were the most popular technologies for 68% SMEs. Only 21% of SMEs used the Internet of Things, such as cars with sensors and smart watches. Approximately 74% SMEs used software to facilitate communication and cooperation, while 81% companies actively used electronic invoicing [49].

Despite the positive trend towards digital digitalization in general, the most essential problem pertains to the small and medium-sized (SMEs) business segment. According to the Latvian Statistical database, SMEs represented 99.8% of all economically active enterprises in 2020 [50]. Moreover, the role of small and medium-sized enterprises (SMEs) tends to increase in Latvia. They represent the backbone of Latvia's economy as key to ensuring economic growth, innovation and job creation. A considerable number of SMEs are still far away from the idea of digital transformation, and the digital skills of employees in many SMEs remain relatively low [51].

In terms of integration of digital technology of businesses, Latvia was ranked 23rd among the EU-27 Member States. The proportion of SMEs having at least a basic level of digital intensity was 42% in Latvia, while the EU average was 60%. Only 9% of Latvian enterprises used big data, 18% used cloud computing services and 11% had web sales to customers; only 7% of revenue for SMEs was generated by e-commerce. It is important to note that an analysis of the use of digital tools by SMEs in each Member State showed low levels of SME digitalization, mainly focusing only on basic digital technologies and not on advanced digital technologies [5].

Among the most important reasons for SMEs adopting digital technologies are: connectivity; online presence; digitalization and automation of business processes; and use of cloud-based services, collaboration and communication. Other specific challenges for SMEs aiming to implement sustainable solutions by leveraging digital technologies include a lack of access to finance for implementing sustainable solutions; lack of knowledge, skills and capacity, particularly with regard to business development; insufficient marketing and strategic management skills; and lack of time [52].

Accordingly, the authors of the paper conclude that for transition to the labour market in the context of challenges of the fourth industrial revolution, it is important for everyone to build up relevant human capital competencies and employability skills, as well as be more successful in implementing the digital transformation in all areas; in particular, this is essential for the small and medium-sized business segment. To achieve this, policy makers need to create new directions in order to manage the transition towards new opportunities successfully. It is important to mention that the future of work will largely depend on the policy decisions countries make. A key challenge is to manage successfully the transition

towards new opportunities for workers, industries and regions affected by the megatrends of technological change and globalization [53].

Based on the above-mentioned problems, the main possible solutions for the successful development of employability skills in Latvia can be ensured by investing in human capital and improving the business environment, the digital transformation of SMEs and the modernization of the education system.

*6.2. Possible Solutions for Successful Development of Employability Skills in Latvia*

The fourth industrial revolution makes changes to the needs of the labour market. The use of new technologies regularly, as never before, depends on the developed human capital, people's professionalism, interest in working actively and being innovative. The transition to a digital society imposes fundamentally much higher requirements for the development of human capital, both to new competencies of specialists and to the process of forming these competencies [54].

One of the hottest debates today is that in 10 years, there will be professions that do not exist today. The question of how to prepare young people for it and how to ensure it is still unanswered. According to Min Xu, Jeanne M. David and Suk Hi Kim, the fourth industrial revolution is more than just technology-driven change. Rather, it is powered by disruptive innovation to positively impact our core industries and sectors, such as education, health and business. In education, with the previous industrial revolutions, the focus of education changed. With the first industrial revolution, education was focused on standard modes of learning. Now, in the fourth industrial revolution, new modes of curriculum and teaching arise, and the focus is changing from modes of teaching to modes of learning. Alternative curriculums are being constantly developed [13].

Education based on innovative future breakthrough technologies increases the "market value" of a specialist in the labour market [55]. Digitalization of education is an integral part of the training of a modern specialist. These trends are associated with a repeated increase in the importance and volume of information, and an increase in the number of interdisciplinary research and projects. If a person has interdisciplinary knowledge, he/she can acquire several other kinds of knowledge, i.e., become "overgrown with knowledge". Interdisciplinary education is exponential. Studies show that students today realize the need to increase their competence in the field of artificial intelligence, processing and analysis of big data, information and communication technologies to successfully enter the labour market. In order to actively support the acquisition of modern knowledge, for example, as part of its global initiative, Microsoft has developed the Digital Skills Development Program (Baltic Digital Skills Development Program) in cooperation with the University of Latvia and other Lithuanian and Estonian universities, which is accessible to everyone. It aims to help individuals to acquire digital knowledge and skills in one of the following areas: business analysis—skills to work with data, such as data visualization and analysis, the basics of cloud computing and its application to business development; data analysis—basics of data visualization, work with the Microsoft Power BI platform and other basic knowledge of data usage; low-code/no code programming—software and system design, as well as development and testing methods, while knowledge of application development will help to simplify, transform and automate business processes that will increase competitiveness in the future job market [56].

As mentioned above, Latvia and EU have a significant and systemic gap between market needs and what is offered in terms of employability skills related to advanced digital technologies. For successful development of employability skills, it is very important to invest in education, as well as to improve the business environment. In this context, the European Commission has created a new Digital Europe Program 2021–2027 [57] with an overall budget of €9.2 billion to shape and support the digital transformation of Europe's societies and economies. The program will boost frontline investments in supercomputing and five key digital sectors: high-performance computing, artificial intelligence, cybersecurity and trust and advanced digital skills, thereby ensuring the wide use and deployment

of digital technologies across the economy and society in order to strengthen European industrial technological leadership.

Similar policy initiatives, starting from a strategic approach to operational implementation, have already taken place in other EU countries. For example, Latvia has developed its Education Development Guidelines 2021–2027 [58], which identifies key policy initiatives that are critical for skills development. However, the document Digital Transformation Guidelines 2021–2027 [59] defines the further development of digitalization as a cross-cutting element for the period 2007–2013, particularly in areas such as innovation and science, education, health, an inclusive society and labour market, infrastructure, regional development, the environment and energy. It is emphasized that the main problem in the way of Latvia's progression towards digital transformation is human capital, as a large segment of the population of Latvia lacks even the so-called basic digital skills, which in turn results in the sluggish transition of companies to digitalization of processes. In order to solve this problem, which is relevant not only in Latvia but also in the world, it is important to make investments in human capital and education rooted in learning how to learn, in providing access to learning opportunities for all, from early childhood education to secondary education, and in strengthening vocational and tertiary education for developing those competences suitable for increasing youth employability. According to the OECD, education is no longer about teaching students something alone; it is more important to be teaching them to develop a reliable compass and the navigation tools to find their own way in a world that is increasingly complex, volatile, and uncertain [60]. To achieve this, it is very important to modernize the education and training systems in order to ensure the link between the educational system and changes in the labour market, as well as the capability to prepare a person for work in changing conditions during his or her whole life.

Therefore, one of the most effective tools for successful development of youth employability skills is their broad involvement in business. In this regard, it is particularly relevant to facilitate collaboration among educators, academics, policymakers and practitioners in order to involve youth in the system of early entrepreneurial activities. To shape the new generation's understanding of entrepreneurship and its desire to engage in, it is very important to balance theory and practice and employ modern teaching methods, giving a greater possibility to students to work practically. In cooperation with entrepreneurs, both professional and academic programs should be enhanced to include student practical work as orders from private firms or government institutions, based on the needs in the national economy, so that the learning outcomes and good ideas are not wasted. Professionals (guest lecturers) from various fields should be more often invited to lectures and seminars, and students have to be offered a possibility to study interdisciplinary programs so that the knowledge and skills acquired could be used in tackling intersectoral problems.

The practical education-based learning approach can also be applied through involving students in the new organizational mechanisms, such as university-based business incubators, technology transfer contact offices and innovation centres, and developing the networks among them. A business incubator is an important tool that can be used by universities to support new start-ups and spin-offs, as well as to build links with industry [60,61]. Working in a business incubator, young people have the opportunity to demonstrate their skills in a specific activity in combination with the acquisition of knowledge: to establish companies, gain experience in working in a team, jointly plan, organize and manage their work, find trade-offs and make decisions independently. By seeking an innovative approach to solving economic problems, they can express themselves creatively, implement their business ideas and develop the ability to take responsibility, initiative, not to be afraid of the unknown, and build confidence that they can achieve what they want.

Thus, collaboration among educators, academics, policymakers and practitioners provides an opportunity to design the learning process so that students can develop their unique abilities, working practically during the academic period, while at the same time acquiring knowledge and developing employability skills. It means that it is essential that

individuals, along with developing specific skills, also build up personal qualities (soft skills), as well as the creation of opportunities for business start-ups being important [62,63].

Overall, the results of the present research show that for successful development of employability skills, it is very important to invest in education, as well as to improve the business environment. This requires new initiatives and more collaboration among educators, academics, policymakers and practitioners in order to involve youth in the system of early entrepreneurial activities. To achieve this, it is very important to modernize the education system, to promote training and lifelong learning in order to ensure the link between the educational system and changes in the labour market. In order to ensure access to a modern learning process and information, digitalization of all schools and libraries of Latvia should be implemented, as well as the capability to prepare a person for work in changing conditions during his or her whole life needs to be developed.

## 7. Conclusions and Future Research Recommendations

### 7.1. Contributions to Theory

From a theoretical point of view, this research expands the literature on the consequences and challenges of the fourth industrial revolution by deepening knowledge about the impact of digital transformation on the labour market, especially employability, identifying the major features of the industrial revolutions and presenting a comparison between them. The research findings show that the fourth industrial revolution is building on the third one, and it is characterized by a fusion of technologies that is blurring the lines between the physical, digital and biological spheres. Schwab [1,17] identified three reasons that the fourth industrial revolution is distinguished from the third revolution: velocity, breadth and depth, and systems impact. In this revolution, emerging technologies and broad-based innovation are diffusing much faster and more widely than in previous ones, which continue to unfold in some parts of the world.

Based on the literature review, the authors identified the positive effect of the challenges of the fourth industrial revolution due to technological innovations, and labour market changes can take place quite rapidly. Automation and robotics technologies taking over simpler, more routine tasks now threaten many professions. As a result, technological innovations tend to raise labour productivity by replacing current workers with technology.

The results of the analysis regarding the research question, "what competencies and skills needed in the future according to technological change?" and the challenges of the labour market show that in the 21st century, the young generation will face the real consequences of the fourth industrial revolution which are changing processes taking place throughout the world; in particular, it is imposing high demands on people's education, their professionalism and for all people to build up employability and digital competences/skills in order to be able to learn and implement new technologies. Based on the literature review, the authors summarize some of the main competencies and skills which strongly relate to employability and are needed in the future according to technological change and the challenges of the labour market. After having studied them, the authors have developed an employability skills framework.

From a theoretical point of view, this research adds to the theoretical framework of employability skills needed in the 21st century, which includes the main elements of employability skills: digital skills, social skills, core skills and contextual skills. Thus, employability is a combination of several factors building up the young individual's key competencies and different skills, including also scientific literacy. The importance of scientific literacy as an element of employability skills nowadays is increasing because science has become part of human daily life today—we use scientific achievements and inventions every day, and it is expected that in the future the importance of scientific literacy of society will only increase.

### 7.2. Contributions to Practice

From a practical point of view, the present research expands and provides insights into the situation in Latvia regarding the impact of the fourth industrial revolution on the development of employability skills and performed an analysis of the main possible solutions that can be ensured by investing in human capital and improving the business environment, the digital transformation of SMEs, the modernization of the education system and promoting training and lifelong learning. This requires new initiatives and more collaboration among educators, academics, policymakers and practitioners in order to modernize the education and training systems and to involve youth in the system of early entrepreneurial activities.

The hierarchy analysis of the development scenarios of young-generation employability in the context of the challenges of the fourth industrial revolution performed allows identification of the most effective scenarios; the first is the scenario "Cooperation between educational institutions and entrepreneurs" (0.39). The second is "The role of policy makers, educational institutions, and entrepreneurs in lifelong learning" (0.28). At the same time, the scenario "Promoting the development of human capital and the business environment" (0.16) had the lowest rating among all the criteria groups. It means that, for successful development of young generation employability, significant attention should be paid to investment in human capital and the digital transformation of business. Investing in human capital and improving the business environment continues to be the best policy to insure against the risk of automation and should be a priority for policy makers. Based on the results of the research, it is concluded that the dynamic changes in the types of work required by the knowledge society pose serious challenges to education systems, as educational institutions are required to prepare young people for jobs that do not currently exist. This indicates that the education system has to focus more on sustainability.

### 7.3. Limitations

Youth employability is affected by a number of factors: the education system, which builds up necessary competences in the youth (skills, knowledge and personal traits); career services, which provide knowledge of the labour market, information on the desired job, the wage or salary, working conditions and career growth opportunities and give a further steer in the direction of employability initiatives; external conditions (macroeconomic situation, labour market demand, labour market legislation); labour market conditions– employers' notion of the desired employee, circumstances to realize one's potential and develop within the job and maintain sustainable employability; as well as government policy measures aimed at promoting employability [55]. Due to the wide scope of the above-mentioned factors, the present research is limited and focuses on an analysis of the education system, employers and government policy aimed at developing youth employability skills in the context of the fourth industrial revolution.

### 7.4. Future Research Recommendations

As regards improving the development of young generation employability according to the consequences of the fourth industrial revolution and the necessity to focus on business digitalization in Latvia in the future, it is advisable to conduct in-depth research into these fields, and researchers must elaborate recommendations to policymakers on the possibilities to make more investment in the development of human capital and to improve youth employability. It is planned to continue this research by using statistical analysis to identify correlations between youth employment opportunities influenced by the fourth industrial revolution.

**Author Contributions:** Conceptualization, V.B. and L.G.; methodology, V.B.; data curation, L.G., B.R. and P.R., software, P.R.; validation, V.B. and P.R.; formal analysis, V.B., L.G., B.R. and P.R.; investigation, V.B., L.G., B.R. and P.R.; writing—original draft preparation, V.B. and L.G.; writing—review and editing, V.B. and L.G.; visualization, P.R.; supervision, V.B. and B.R.; resources, V.B., L.G., B.R. and P.R., project administration, B.R.; funding acquisition, B.R. All authors have read and agreed to the published version of the manuscript.

**Funding:** This research was funded by Latvia Ministry of Education and Science, State research programme "Challenges for the Latvian State and Society and the Solutions in International Context (INTERFRAME-LV)", grant number VPP-IZM-2018/1-0005.

**Institutional Review Board Statement:** Not applicable.

**Informed Consent Statement:** Not applicable.

**Data Availability Statement:** The data for this study were obtained from the following databases: OECD data https://read.oecd-ilibrary.org/employment/oecd-employment-outlook-2019_9ee00155 -en#page1 (accessed on on 2 November 2021); UNESCO https://unesdoc.unesco.org/ark:/48223/pf0 000379707 (accessed on 2 March 2022); European Commission. https://digital-strategy.ec.europa.eu/ en/policies/desi (accessed on 2 March 2022); Baltic CFO survey https://www.seb.lv/sites/default/ files/2021-10/2021_Baltic_CFO_survey.pdf?msckid=494ad4a1b65c11ec909583127574ad68 (accessed on 10 December 2021).

**Conflicts of Interest:** The authors declare no conflict of interest.

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
