# Peer review of "Consequences and Challenges of the Fourth Industrial Revolution and the Impact on the Development of Employability Skills"

_sustainability, doi:10.3390/su14126970_

Round 1
Reviewer 1 Report
After the revision, the paper can be published.
Author Response
Dear Sir/ Madam,
We express our deep gratitude for reviewing this article.
With best regards,
Liva Grinevica
Veronika Bikse
Baiba Rivza
Peteris Rivza
Reviewer 2 Report
As attachment.

Author Response

(The authors gave the same response as above.)

Reviewer 3 Report
The reviewed revised paper focus on an interesting topic. The scientific quality of the revised paper is averagre / up to above average. The authors are required make to deeper theoretical analysis in broader framework. I recommend add to discussion issues about knowledge, knowledge sharing, emotional intelligence. They are very important for development of employability skills. Some authors write about this issues, I can recommend for example this:
Bencsik, A. et al. 2019. Impact of Informal Knowledge Sharing for Organizational Operation. Entrepreneurial Business and Economics Review, 7(3), 25-42.
Mura, L., Zsigmond, T., & Machová, R. 2021. The effects of emotional intelligence and ethics of SME employees on knowledge sharing in Central-European countries. Oeconomia Copernicana, 12(4), 907–934.
Author Response
Dear Sir/ Madam,
We express our deep gratitude for reviewing this article. We have checked the references throughout the article, everything is correct, and the references correspond to the cited articles, the references correctly indicate the sources used and the authors' articles.
We believe that the idea of ​​the reviewer's 3 suggestion is a good, but new research and analysis is needed to make such an addition. We could take this suggestion into account in future articles.
With best regards,
Liva Grinevica
Veronika Bikse
Baiba Rivza
Peteris Rivza
This manuscript is a resubmission of an earlier submission. The following is a list of the peer review reports and author responses from that submission.
Round 1
Reviewer 1 Report
Thank your for the opportunity to review the manuscript.
This manuscript summarized the consequences and challenges of the fourth industrial revolution and the impact on the development of employability skills. The research questions proposed in the manuscript are of practical significance. However, I have three major concerns about this research。
- The study investigated the impact of the 4th industrial revolution on labor market, especially the employability. However, when introducing the background of the 4th industrial revolution, the study did not give a systematic comparison between the typical features of this revolution and those of other industrial revolutions. I would suggest the author to make a comparison and then summarize the differences, and then to come up with the opportunities and challenges based on the systematic comparison and summary.
- The study suggested that the 4th industrial revolution requires the development of employability skills. However, there is not enough evidence to support the argument. Specifically, what are the industries that need enhanced development of employability skills?Certainly, different industries require different skills and talents. Without goes into details, the study might seem to be talking about this in general.
- The study put forward four scenarios for improving the integration of young people into the labor market. Specifically, based on what theory and logic does the study constructed the four scenarios? I think the study might need to elaborate on this point.
- As the contribution to theory, the study needs to make clear what the present study contributes to the theory in comparison with its extant literature. For example, extant literature on employability skills might have already discussed the importance of such skills as communication, collaboration, creativity, etc. Therefore, what does the present study add to the theory?
Reviewer 2 Report
The main question of the research is to perform an analysis of the consequences and the impact of the fourth industrial revolution on the development of employability skills and to identify 15 the possible solutions to help overcome these challenges. The topic is original and actual. It addresses the problem with regard to the implementation of advanced technologies was a lack of qualified specialists and the fact that a large segment of the population lacked digital skills.
The paper is well written and the text is easy to read.
There are some suggestions for improvement:
Please, expand the chapter Methodology. Form the part of the paper Methodology with the sub-chapters: i.e. Sample (participants), research questions, procedure ...
The conclusions are consistent with the evidence and arguments presented. Please, add answers to the research questions at the Conclusion.
Reviewer 3 Report
The submitted paper brings an interesting topic, but the research design, methodology is very poor. Any hypotheses, any deeply research approaches. The quality of the reviewed paper is not at the adequate level.
Reviewer 4 Report
The use of the Analytic Hierarchy Process (AHP) methodology is an adequate targeting of the article, the content analysis could be performed with a wider range of relevant resources available on the issue outside the European Union, although this context is indisputably decisive. The analysis was performed as standard and the results are conclusive and logically argued. Formally, the article is suitably structured, it is clear, the sources are correctly cited and listed in the References.
Reviewer 5 Report
As attachment
